# Animal Models for Studying Protein-Bound Uremic Toxin Removal—A Systematic Review

**DOI:** 10.3390/ijms241713197

**Published:** 2023-08-25

**Authors:** Sabbir Ahmed, Joost C. de Vries, Jingyi Lu, Milan H. Verrijn Stuart, Silvia M. Mihăilă, Robin W. M. Vernooij, Rosalinde Masereeuw, Karin G. F. Gerritsen

**Affiliations:** 1Division of Pharmacology, Utrecht Institute for Pharmaceutical Sciences, Universiteitsweg 99, 3584 CG Utrecht, The Netherlands; s.ahmed@uu.nl (S.A.); j.lu@umcg.nl (J.L.); s.mihaila@uu.nl (S.M.M.); r.masereeuw@uu.nl (R.M.); 2Department of Nephrology and Hypertension, University Medical Center Utrecht, 3584 CX Utrecht, The Netherlands; j.c.devries-34@umcutrecht.nl (J.C.d.V.); m.h.verrijnstuart@students.uu.nl (M.H.V.S.); r.w.m.vernooij-2@umcutrecht.nl (R.W.M.V.); 3Julius Center for Health Sciences and Primary Care, University Medical Center Utrecht, Utrecht University, 3584 CX Utrecht, The Netherlands

**Keywords:** chronic kidney disease, protein-bound uremic toxins, animal models, indoxyl sulfate, hippuric acid, para-cresyl sulfate

## Abstract

Protein-bound uremic toxins (PBUTs) are associated with the progression of chronic kidney disease (CKD) and its associated morbidity and mortality. The conventional dialysis techniques are unable to efficiently remove PBUTs due to their plasma protein binding. Therefore, novel approaches are being developed, but these require validation in animals before clinical trials can begin. We conducted a systematic review to document PBUT concentrations in various models and species. The search strategy returned 1163 results for which abstracts were screened, resulting in 65 full-text papers for data extraction (rats (*n* = 41), mice (*n* = 17), dogs (*n* = 3), cats (*n* = 4), goats (*n* = 1), and pigs (*n* = 1)). We performed descriptive and comparative analyses on indoxyl sulfate (IS) concentrations in rats and mice. The data on large animals and on other PBUTs were too heterogeneous for pooled analysis. Most rodent studies reported mean uremic concentrations of plasma IS close to or within the range of those during kidney failure in humans, with the highest in tubular injury models in rats. Compared to nephron loss models in rats, a greater rise in plasma IS compared to creatinine was found in tubular injury models, suggesting tubular secretion was more affected than glomerular filtration. In summary, tubular injury rat models may be most relevant for the in vivo validation of novel PBUT-lowering strategies for kidney failure in humans.

## 1. Introduction

Proximal tubular secretion represents an essential kidney function capable of rapidly clearing many exo- and endogenous compounds, including protein-bound uremic toxins (PBUTs). These toxins are minimally filtered by the glomerulus due to their extensive binding to plasma proteins, in particular albumin. In physiological settings, PBUTs are actively secreted into the urine by proximal tubular cells via a concerted action of influx transporters at the basolateral membrane (viz. organic anion transporter 1 (OAT1/SLC22A6) and OAT3 (SLC22A8)) and efflux transporters at the apical membrane (viz. multidrug resistance proteins (MRP4/ABCC4) and breast cancer resistance protein (BCRP/ABCG2)) [1,2,3]. Progressive kidney disease often leads to a decline in tubular (transport) function affecting PBUTs clearance, which results in PBUTs accumulation in plasma [4,5,6].

Many PBUTs have been linked to the pathogenesis of comorbidities in chronic kidney disease (CKD), in particular cardiovascular disease, to the progression of CKD, and to its associated mortality [7,8,9,10,11,12]. Evidence from in vitro studies showed, amongst others, that PBUTs induce oxidative stress, inflammation, and fibrosis in cultured proximal tubular epithelial cells [13,14], endothelial dysfunction by ROS production [15,16,17], and vascular calcification by activating the NF-κB signaling pathway [18]. Preclinical in vivo studies suggested that PBUTs may play a crucial role in the progression of CKD via the activation of the renin-angiotensin-aldosterone system/TGF-β1 pathway [19] and kidney tubular cell damage [13], and may promote cardiovascular diseases via vascular calcification [20,21,22], cardiac fibrosis [23], and inflammation pathways [22]. Various clinical studies confirmed the adverse effects of PBUTs in this context. In a large prospective cohort study evaluating 3416 CKD patients, Chen et al. found that reduced renal clearance of PBUTs is significantly associated with the progression of CKD and all-cause mortality [24]. Some clinical studies demonstrated that the plasma concentration of PBUTs is related to CKD progression (dependent or independent of eGFR) [4,5]. A prospective study of 200 CKD patients reported that urinary excretion of para-cresyl sulfate (pCS) is a predictor of cardiovascular events in CKD, independent of eGFR [25]. Thus, reducing PBUT accumulation seems to be important in the treatment of patients with CKD. However, dialysis treatment for kidney failure only removes the free fraction of PBUTs, which results in a low total PBUT removal. Indeed, in dialysis patients, the concentration of endogenous secretory solutes was shown to be substantially more increased than that of filtration solutes [26]. For example, a conventional hemodialysis session showed a reduction ration of 80% for urea, which is a non-protein-bound small molecule, and a reduction ratio of only <35% for indoxyl sulfate (IS) and pCS [27,28]. Currently, various innovative strategies and concepts are under development that target the removal of PBUTs in kidney failure [29]. These include (i) bioartificial kidney (BAK) devices equipped with proximal tubule epithelial cells with transporters capable of active secretion [30,31], (ii) adsorption techniques such as mixed matrix membranes that contain activated carbon capable of adsorbing PBUTs efficiently [32], and (iii) techniques aimed at displacing PBUTs from their binding protein to enhance filtration, such as the modification of osmotic conditions within the dialyzer by hypertonic predilution, the application of electromagnetic fields or the use of albumin-binding competitors [33,34,35]. However, prior to clinical evaluation, the safety and effectiveness of these innovations may ideally be evaluated in animals. For this, reliable uremic animal models are required with PBUT concentrations within the range of those in human kidney failure. This review aims to provide an overview of PBUTs concentrations in different species and uremic models with two distinct types of kidney damage, either ‘nephron loss’ (such as in nephrectomy models) or tubular damage. Since PBUTs are primarily cleared by active tubular secretion, we hypothesized that tubular damage models would show higher plasma PBUTs concentrations than nephrectomy models. To this end, we categorized PBUTs data by the type of kidney damage inflicted.

## 2. Materials and Methods

### 2.1. Literature Search and Study Selection

A literature search was conducted for eligible studies until September 2022, using Pubmed and Embase with the following terms: kidney diseases, chronic kidney disease, uremic toxin, protein-bound uremic toxin, and animals. The complete search string is available in the Appendix A. In addition, reference lists of published narrative reviews were checked for eligible studies. We only included studies reporting plasma or serum concentrations of PBUTs and at least one conventional filtration marker (e.g., creatinine (Cr) or urea) in animals with impaired kidney function. Studies on genetically altered animals were also included if the wild-type group was included, and only the PBUTs concentrations in the wild type were extracted from these studies. Reviews, conference abstracts, and studies not reported in the English language were excluded. Eligible studies were selected independently either by SA or JV.

### 2.2. Data Extraction

All relevant data were extracted from the papers by one researcher (SA), after which the data were independently validated and curated by a second researcher (JV). The data included plasma concentrations and urinary excretion of PBUTs, creatinine, or urea, as well as induction method of kidney injury, follow-up time, and body weight. Data from graphs were extracted using GetData Graph Digitizer v2.26. Where necessary, standard errors of the mean (SEM) were converted to standard deviations (SDs), concentrations of Cr and PBUTs were converted to µmol/L, and those of urea to mmol/L. Finally, if the number of subjects in a group was unclear, but a range of subjects was given (e.g., “data shown are from 6 to 8 animals”), the lowest number was used in the analyses as this would result in an underestimation of the true effect.

### 2.3. Analysis

A descriptive analysis of the data was performed using one-sided forest plots with a random effect model—stratified by damage type and tested for subgroup difference with separate plots for healthy and diseased—of the plasma Cr, urea, IS, pCS and hippuric acid (HA) concentrations. Subtotal nephrectomy and cisplatin/adenine treatment models were considered nephron loss and tubular damage models, respectively. The ratio of the mean plasma concentration in uremic animals to that in healthy animals (fold change) was calculated per study for IS, Cr, and urea and depicted in one-sided forest plots. The ratio of the fold change in IS to that of Cr was calculated for each study and depicted as dot plots. Furthermore, the influence of strain difference on the IS, Cr, and urea concentrations in rats was evaluated using one-sided forest plots with separate plots for healthy and diseased.

Studies not reporting both healthy and uremic concentrations or number of animals per group were not included in the forest plots.

Data were analyzed using R studio v4.0.5 with ‘ggplot2′ and ‘meta’ plugins (one-sided forest plots). The dataset and scripts are available in the Appendix A. Additional plots were made using GraphPad Prism v9.3.0. A two-sided *p*-value of <0.05 was considered statistically significant. All data are displayed as mean ± SD, unless stated otherwise.

## 3. Results

### 3.1. Characteristics of the Studies

The study selection process is shown in a flow chart (Figure 1). A total of 1163 records were retrieved from the electronic search, of which 63 articles met the inclusion criteria. Additional screening of the reference lists in published narrative reviews from other sources resulted in the inclusion of two additional articles. Therefore, a total of 65 original research articles were used for the extraction of data. Among them, 41 reported PBUTs in uremic rats, 17 in mice, 3 in dogs, 4 in cats (2 studies reported both cats and dogs), 1 in goats, and 1 in pigs (Appendix A). Only rat and mouse studies reporting IS were included in further analyses, as data were too limited for the studies on larger animals and other PBUTs. Of the 41 rat studies, 3 were excluded from the analysis due to missing plasma concentrations for healthy animals. We identified 22 studies employing a nephron loss model (i.e., subtotal nephrectomy), and 15 tubular damage models (cisplatin- or adenine-induced) suitable for analysis in a forest plot. Among the mice studies, 8 studies with a nephron loss model and 7 with a tubular damage model were included for analysis.

### 3.2. Overview of PBUTs in Healthy vs. Uremic Animals

The ranges of mean or median plasma concentration of IS, pCS, HA, and Cr in different species are depicted in Table 1. Overall, the ranges of PBUT concentration were broad in all species, where data extracted from dog and cat studies were highly heterogeneous.

In rodents, overall plasma PBUT concentrations showed an increase after kidney damage, which was relatively higher than that of conventional markers (i.e., Cr and urea) (Figure 2, Figure 3, Figure 4 and Figure 5, Appendix A). The rat studies showed a weighted average of plasma IS (Figure 2) that was 11.2-fold higher in uremic compared to healthy animals, whereas plasma Cr (Appendix A) and urea (Appendix A) were 3.6 and 4.9 times higher, and Cr clearance (Appendix A) 3.6 times lower, respectively. Based on the fewer number of available studies, the weighted average of plasma pCS (Figure 3) and hippuric acid (HA; Figure 4) were 22.2- and 7.6-fold higher in uremic than in healthy animals. Compared to rats, the overall changes in these markers were less pronounced in uremic mice. The weighted average of plasma IS (Figure 5), Cr (Appendix A), and urea (Appendix A) were 6.4, 2.8, and 3.3 times higher in uremic than in healthy mice.

### 3.3. Comparison among Injury Types and Strains

The extent of PBUT accumulation may vary among animal models with distinct sites of injury since their clearance largely depends on the secretory function, which is restricted to the proximal tubule. To understand the influence of injury site, the extent of PBUTs accumulation between nephron loss and tubular injury models was compared. In rats, most studies with tubular injury models showed higher plasma IS concentrations than those with nephron loss models, which amounted to a 3.6-fold higher weighted average in tubular injury studies. However, this was not observed in mice. We next calculated the fold change (ratio of mean concentration in uremic to that in healthy animals) of IS (Appendix A) and Cr (Appendix A) in individual studies to exclude the variability at baseline among studies. Subsequently, the ratios of IS fold change to Cr fold change were calculated and depicted in dot plots (Figure 6). In rat studies, these ratios were higher in tubular injury models compared to nephron loss models. This was not the case for mice, where only few ratios could be calculated (Figure 6). Notably, the follow-up period for most studies with tubular injury models was shorter than that in nephron loss models in both rats and mice (Appendix A).

Furthermore, to evaluate the variability among different strains, we compared the two most frequently used rat strains, viz. Sprague–Dawley and Wistar. Since only two nephron loss studies used Wistar rats, we focused on tubular injury models for comparison. There was no difference in IS concentrations between these strains. In the studies with Sprague–Dawley rats, the elevation of IS was more consistent but with a higher variation than those in Wistar rats (Appendix A). For plasma creatinine and urea, no differences were observed between the strains (Appendix A, respectively).

## 4. Discussion

The goal of this study was to provide an overview of PBUTs concentrations in different species and uremic models which can be used to select suitable animal models for in vivo preclinical testing of novel methods for removing PBUTs from patients with impaired renal function. Overall, the plasma concentrations of PBUTs extended over a broad range. In rats, the studies on tubular injury models documented higher mean uremic plasma IS, more often within the range found in human kidney failure, than the studies on nephron loss models. When compared to the increase in plasma Cr in the uremic situation, the (relative) increase in plasma IS was more pronounced in tubular injury than in nephron loss rat models.

In healthy conditions, most of the rodent studies showed plasma IS concentrations above the healthy human range while, for pCS and HA, most plasma concentrations were within or below the healthy human range. Among the larger animals, mean IS concentration in goats was within the human range, whereas dogs and cats had much higher, and pigs had lower concentrations. The variability in plasma PBUTs concentrations among studies and species may be influenced by different factors. The endogenous production of PBUTs involves colonic microbial fermentation of the dietary protein-derived amino acids, tryptophan and tyrosine, into metabolites that undergo further biotransformation, such as sulfation and hydrogenation, by enzymes in the hepatic portal system [38,39]. Therefore, the plasma concentration of PBUTs can be influenced by, for instance, variations in gut microbiota, dietary protein intake, and hepatic metabolism of nonpolar precursors such as indole [40]. Furthermore, interspecies variability may contribute to differences in endogenous metabolism and colonic microbiome [41,42,43]. In addition, a longer colonic emptying time may favor increased PBUT production, as evidenced by a significant correlation between longer colonic transit time and higher urinary excretion rates of phenols [38,44]. Compared to humans, the uptake of food in rodents is more frequent, which may lead to a continuous exposure of the microbiome to intestinal contents, increased microbial fermentation, and increased intestinal absorption of the precursors of PBUTs. This might explain the higher concentration of plasma IS in healthy rodents than in humans [45]. This was not evident for HA and pCS, possibly due to the fewer number of studies. Unfortunately, the data were too limited to investigate the association between PBUT concentrations and interspecies or inter-strain differences in the gut microbiome.

In uremic conditions, most rodent studies documented a mean plasma IS concentration within the human uremic range (175 ± 125 µM [37]), while the data from a small number of studies using large animals and those reporting other PBUTs were very heterogeneous. Unlike IS, the uremic plasma pCS and HA concentrations in most rat studies were much lower than the human range. The studies with dogs, cats, and pigs showed a lower uremic concentration of PBUTs than in humans, except for one study in dogs that reported very high concentrations of plasma IS [46]. PBUTs concentrations in goats were comparable to the human uremic range. These variations in uremic PBUT concentrations between and within species may be attributed to various factors, including differences in dietary protein intake, hepatic metabolism, and the colonic microbiome as discussed above. Additionally, in uremic conditions, the composition and function of the gut microbiome may undergo a substantial change leading to the altered metabolism of colonic bacteria and associated metabolites [39,47,48]. The altered gut environment may also influence PBUT production. In human CKD, altered protein binding was shown to interfere with renal clearance of PBUTs, including IS, possibly due to post-translational modifications such as glycosylation and oxidation [49,50].

The site of injury was found to affect the plasma accumulation of IS in rats. Compared to nephron loss models, studies with tubular injury models showed higher mean plasma IS concentrations. Standard deviations reflecting interindividual variability were lower in the nephron loss than in the tubular injury studies, despite applying complex surgeries and longer follow up periods. We further compared the ratio of fold change in plasma IS to the fold change in Cr for individual studies, which was significantly higher in rats with tubular injury than in the models with nephron loss. This can be explained by the fact that in tubular injury models the tubular function, which includes active proximal tubular PBUT secretion and is relatively more affected than the glomerular filtration function while, in nephron loss models, both functions are equally affected. In addition, the tubular injury models may better reflect the primary pathophysiological mechanism behind PBUT accumulation in human CKD, i.e., proximal tubular dysfunction or drug-induced nephrotoxicity. Also, the structural changes causing decreased PBUT clearance in tubular injury models resemble those in human CKD (viz. primarily interstitial fibrosis and tubular atrophy) [51,52]. Adenine-fed rats showed structural damages, such as tubular epithelial fibrosis, flattening of tubular epithelial brush borders, crystal deposition in the tubulointerstitium, and tubular epithelial hypertrophy [51,53]. Similarly, cisplatin induced structural changes such as fibrosis, necrosis, and apoptosis of the tubular epithelial cells and loss of brush border were reported [52,54,55,56]. The nephrectomy models, on the other hand, were generated by removing nephron mass by surgical excision or by ligation of renal arteries. In the remnant kidneys of rodent nephrectomy models, both glomerulosclerosis and tubulointerstitial fibrosis and tubular atrophy were observed, thus damaging both the glomerular and tubular apparatus and affecting both filtration and secretion function [51,57,58].

In contrast to rats, the ratio of fold change in plasma IS to Cr was not significantly different in tubular injury than in nephron loss models in mice, suggesting that the tubular injury models included in our investigation did not experience more tubular than glomerular damage. Although this might be attributed to the limited sample size, the vulnerability of the mice strains to tubular damage may have also played a role. Recent studies suggest that different animal strains within the same species show variability in disease susceptibility, which may influence the results [59,60]. For instance, following kidney injury, voluntary wheel running exacerbated GFR decline in Sprague–Dawley but not in Fisher rats [61]. B6J mice were reported to be relatively resistant to cisplatin-induced tubulointerstitial fibrosis, whereas FVB/n mice are more prone to such damages and may be a better model for studying cisplatin-induced nephrotoxicity [60]. Furthermore, B6J showed higher resistance to kidney fibrosis, hypertension, and albuminuria in various experimental models than other mouse strains such as Balb/C [62,63,64]. In our data, most of the studies used C57BL/6 or its sub-strain C57BL6/J mice. The lack of a significant increase in the ratio of fold change in plasma IS to Cr may be related to the lower susceptibility of this strain to kidney tubular injury. Since only two studies used other strains, we were unable to assess differences in PBUT accumulation amongst mice strains. We could only evaluate the difference between Sprague–Dawley and Wistar rats in tubular injury models, as the number of reports on other models/species were not sufficient for analysis (e.g., only two published studies with nephron loss model used Wistar rats). The studies with Sprague–Dawley rats showed relatively lower inter-study variability in uremic IS concentration with a weighted average closer to that of humans (EUTOX) [37]. However, intra-study standard deviation was lower in Wistar rats.

It is important to note that the selection of a suitable animal model requires additional considerations to be taken into account, such as feasibility of the injury method, mortality rate, and post injury/surgery maintenance. Nephrectomy models involve complex surgical procedures, often multiple, which can lead to a high mortality rate. On the other hand, tubular injury can be induced without surgery, for instance, by administrating a specific percentage of adenine in diet of the animals [51]. The costs and availability of animal housing and trained personnel may limit the feasibility of utilizing a long-term follow-up model for CKD, and instead rely on well-established AKI models. The scarcity of studies involving large animals can be attributed to various factors, including higher expenses for housing and maintenance per animal, laborious surgical procedures, and lack of technical expertise [65].

During analyses, we encountered several reporting-related limitations. Notably, relevant parameters, such as sample sizes, animal characteristics (sex, weight, age), and animal handling (e.g., diet), were not always reported. Furthermore, a number of studies did not report (relevant) data for a control group, which—as with missing sample size—made the inclusion of these reports in the statistical analysis not possible. The analysis was limited to only a few parameters due to the low availability of other parameters, in particular PBUTs (other than IS, pCS and HA) and urinary data. Most studies only reported the absolute daily urinary secretion of PBUTs (Appendix A). Another limitation is that our study was unable to analyze sex-based differences in PBUTs. Sex is known to be a significant factor in animal models that may affect kidney pathophysiology. For example, male mice may be more prone to developing tubular damage upon being treated with adenine [66]. Most rodent studies included in our analysis used males, except for two rat studies [67,68]. Among mouse studies, one [39] did not report sex, one reported both sexes [69], and one reported females, but without information on plasma/serum IS data [70]. Finally, the risk of bias assessment was not performed as it is not designed with this type of (systematic) review in mind, and thus would not accurately reflect the overall quality and validity of these PBUT measurements.

## 5. Conclusions

Most rodent tubular injury models showed a mean IS concentration within the human uremic range. Additionally, plasma IS accumulation was more pronounced in tubular injury than in nephron loss models in rats, and also when normalized for creatinine increase, suggesting that tubular secretory function was more affected than glomerular filtration function. Thus, tubular injury rat models may be the most pragmatic for studying PBUT removal. Data on other PBUT concentrations, or on large species, were limited. To our knowledge, this is the first study summarizing PBUT concentrations in animals and providing comparative evaluations that may be useful in selecting animal models for the in vivo preclinical testing of novel PBUT-lowering technologies.

## Figures and Tables

**Figure 1 ijms-24-13197-f001:**
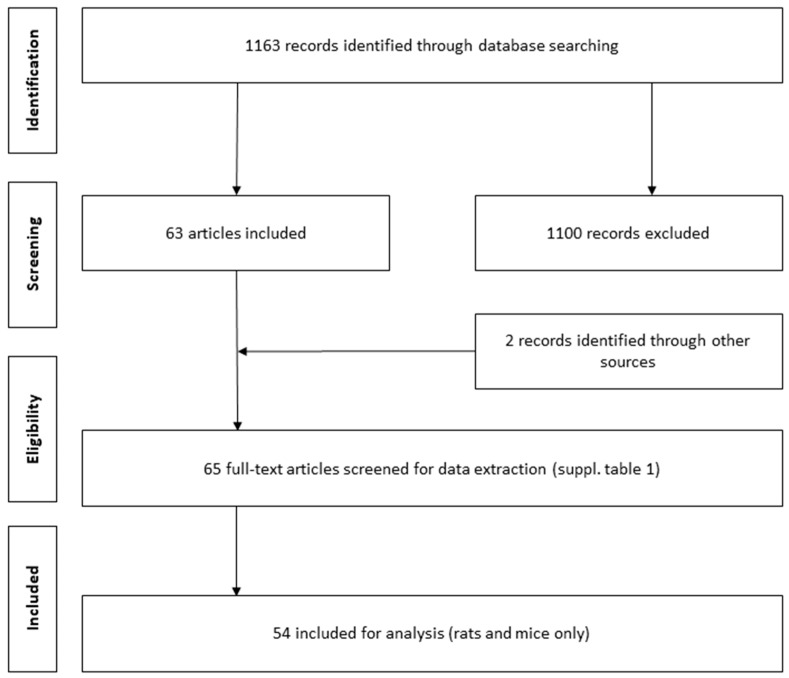
Flow diagram of the article inclusion process. This process was performed by two independent reviewers. Only studies reporting plasma or serum concentrations of PBUTs and at least one conventional filtration marker (e.g., creatinine or urea) in animals with kidney disease were screened.

**Figure 2 ijms-24-13197-f002:**
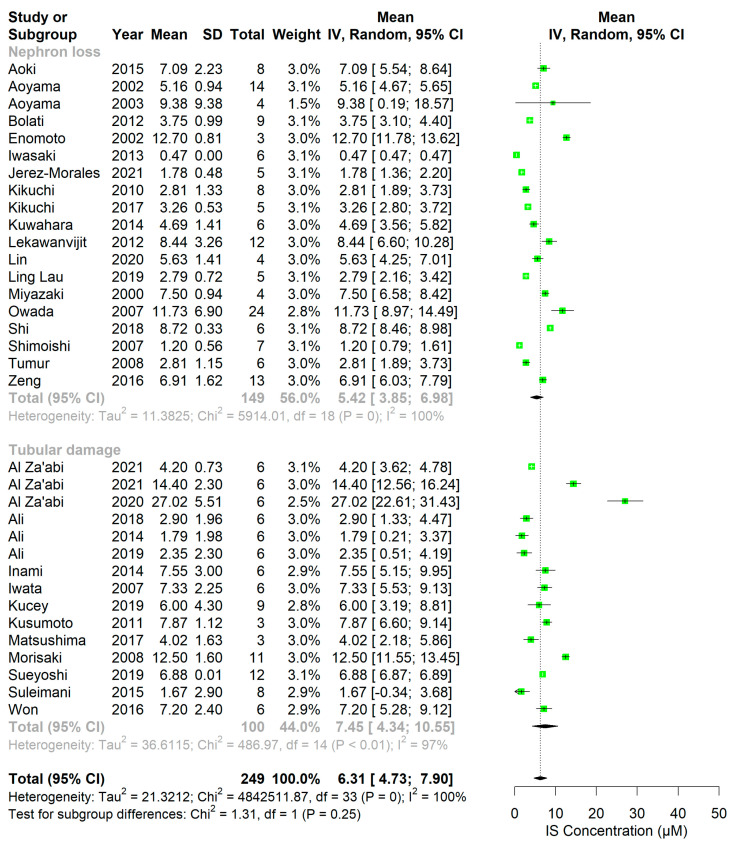
One-sided forest plots of indoxyl sulfate concentrations (IS; in µM) in healthy (**top**) and diseased (**bottom**) rats stratified by damage type. Notably, a different scale range is used for healthy and diseased animals.

**Figure 3 ijms-24-13197-f003:**
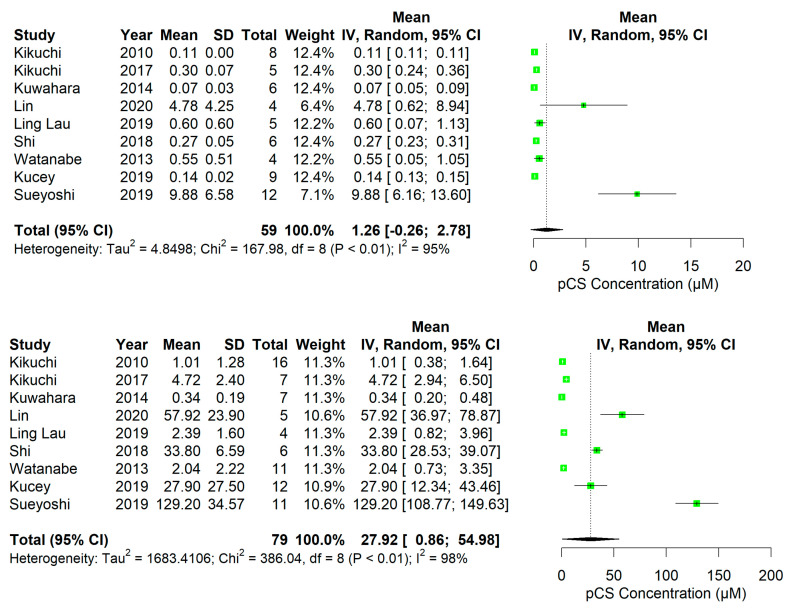
One-sided forest plots of p-cresyl sulfate (pCS; µM) in healthy (**top**) and diseased (**bottom**) rats. Notably, a different scale range is used for healthy and diseased animals.

**Figure 4 ijms-24-13197-f004:**
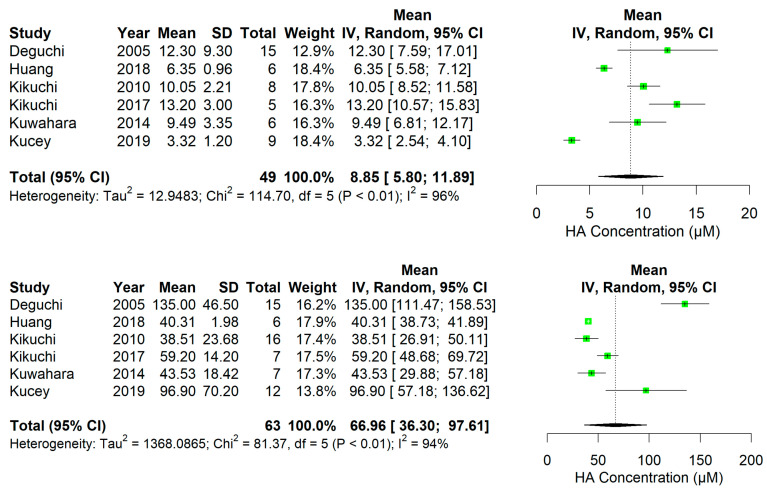
One-sided forest plots of hippuric acid (HA; µM) in healthy (**top**) and diseased (**bottom**) rats. Notably, a different scale range is used for healthy and diseased animals.

**Figure 5 ijms-24-13197-f005:**
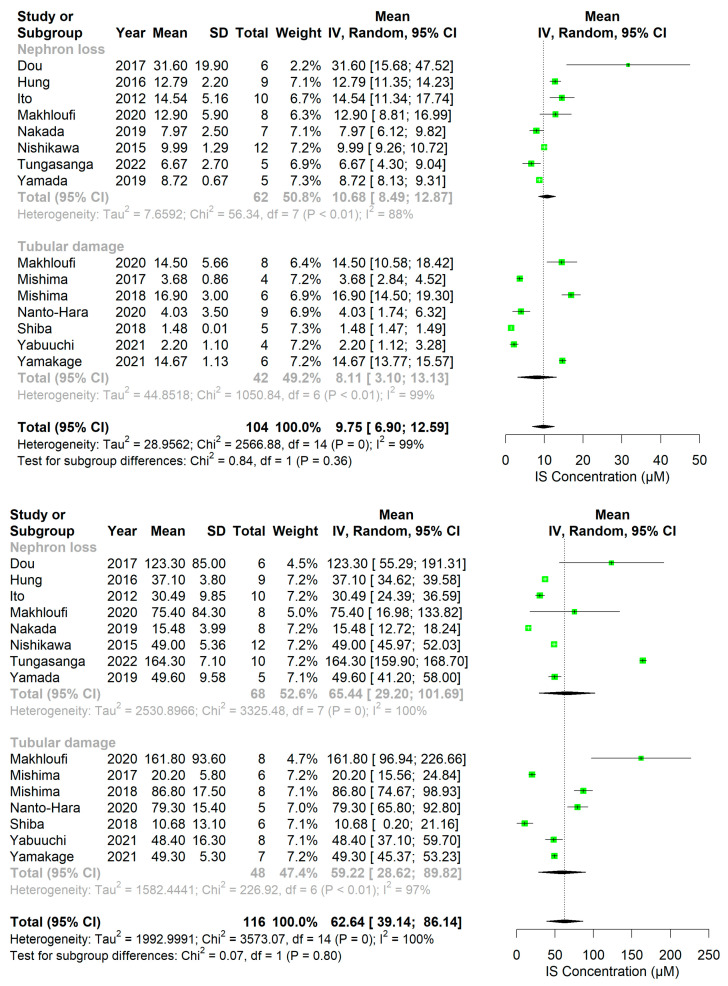
One-sided Forest plots of indoxyl sulfate concentrations (IS; in µM) in healthy (**top**) and diseased (**bottom**) mice stratified by damage type. Notably, a different scale range is used for healthy and diseased animals.

**Figure 6 ijms-24-13197-f006:**
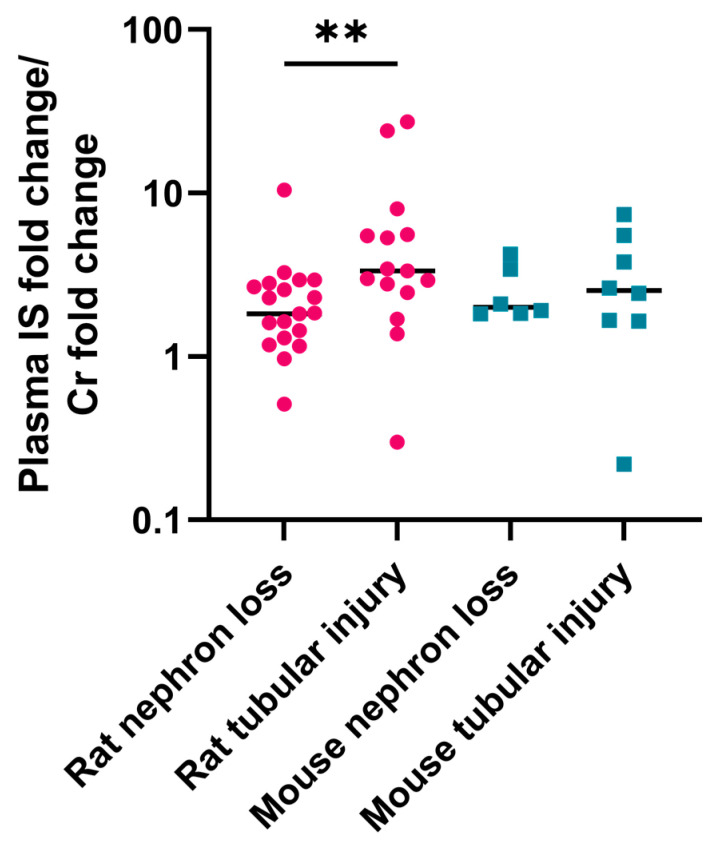
The ratio of indoxyl sulfate (IS) fold change to creatinine (Cr) fold change in rats (red) and mice (blue). Each dot represents one individual study. Non-parametric Mann–Whitney test was performed for statistical analysis. ** *p* < 0.01.

**Table 1 ijms-24-13197-t001:** Summary of plasma IS, pCS, HA, and Cr concentrations in different species. See Appendix A for details.

Species	IS (µM)		pCS (µM)		HA (µM)		Cr (µM)	
	Healthy	Uremic	Healthy	Uremic	Healthy	Uremic	Healthy	Uremic
Rat ^	0.47–27.0 (*n* = 249)	2.47–200 (*n* = 255)	0.07–9.88 (*n* = 59)	0.34–129 (*n* = 79)	3.32–13.2 (*n* = 49)	38.5–135 (*n* = 63)	13.2–88.4 (*n* = 285)	48.3–208 (*n* = 291)
Mouse ^	1.48–31.6 (*n* = 104)	10.7–164 (*n* = 116)	0.86–19.0 (*n* = 24)	0.72–140 (*n* = 40)	2.44 ± 3.22 (*n* = 9) ^#^	25.7 ± 5.81 (*n* = 5) ^#^	1.31–58.5 (*n* = 81)	9.31–234 (*n* = 92)
Dog	33.8 ± 41.3 (*n* = 63) ^$^	83.0–8348 (*n* = 168) *	NA	NA	25.67 ± 2.23 (*n* = 5) ^#^	213 ± 130 (*n* = 5) ^#^	106 ± 53 (*n* = 63) ^$^	186–743 (*n* = 168) *
Cat	5.64–69.4 (*n* = 47) *	13.7–121 (*n* = 230) *	15.4 (4.79–38.36) (*n* = 11) ^@^	28.2–36.9 (*n* = 30) *	NA	NA	106–133 (*n* = 47) *	177–570 (*n* = 230) *
Goat ^#^	1.39 ± 0.50 (*n* = 5)	69 ± 76 (*n* = 11)	37.0 ± 18.0 (*n* = 5)	985 ± 843 (*n* = 11)	43.0 ± 20.0 (*n* = 5)	817 ± 913 (*n* = 11)	65.0 ± 5.6 (*n* = 5)	841 ± 584 (*n* = 11)
Pig ^#^	0.55 ± 0.33 (*n* = 5)	26.1 ± 27.7 (*n* = 5)	0.24 ± 0.26 (*n* = 5)	4.85 ± 5.26 (*n* = 5)	10.1 ± 4.15 (*n* = 5)	63.8 ± 99 (*n* = 5)	103 ± 14 (*n* = 5)	932 ± 470 (*n* = 5)
Human [36,37]	1.13–3.86	46.9–280	3.2–17.0	44.2–289	5.6–27.9	321–831	85.57 ± 2.12	1202 ± 407

^ range of means; * range of medians; ^#^ one paper with mean ± SD; ^$^ one paper with median ± IQR; ^@^ one paper with median and range; IS = indoxyl sulfate; pCS = p-cresyl sulfate; HA = hippuric acid; Cr = creatinine; *n* = number of animals.

## Data Availability

The dataset and scripts are available in the Appendix A.

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
