# Peer review of "Animal Models for Studying Protein-Bound Uremic Toxin Removal—A Systematic Review"

_ijms, 2023, doi:10.3390/ijms241713197_

Round 1

Reviewer 1 Report

Dear Authors, I have read with interest your manuscript. The paper addresses an very interesting issue regarding uremic toxins which interact negatively with biological functions. More than that, protein bound toxins are implicated not only in the progression of chronic kidney disease (CKD), but also in the generation and aggravation of cardiovascular disease.

I would like to address a few suggestions/ questions:

I suggest usind the term ‘’colonic microbial metabolism’’ while searching database for eligible studies.

 When comparing the two most used rat strains, I suggest making correlation between protein bound uremic toxins, urea, creatinine and gut microbioma, knowing the microbioma influence on protein bound uremic toxins.

  This topic is very interesting to discus becausethese theories launch new horizons in chronic kidney disease research.

Author Response

Dear Reviewer 1,
We would like to thank you for reviewing our manuscript. Please find attached a Word file with our response.

Kind regards.

Reviewer 2 Report

Overall impressions

The concept of finding the most suitable animal model to test a novel therapy is a logical step and the general approach that the authors have taken here is in keeping with this concept.

The authors could have made more of the conclusion. Whilst acknowledging the poor quality of evidence available, it would appear that the data from rat tubular injury models would appear to be the most pragmatic.

Introduction

Line 23 and subsequent – ESKD : ideally use the term Kidney failure based on KDIGO nomenclature as per Levey AS et al. Nomenclature for kidney function and disease: report of a Kidney Disease Improving Global Outcomes (KDIGO) consensus conference. Kidney Int 2020; 97: 1117-29.

Line 43: a.o. – what does this abbreviation mean?

Results

Fig 2-5 – visually confusing because the scales for both healthy and diseased groups are different. The visual effect would have been easier if the same scale had been used for both groups. Might have been better off providing overall effect plots for each group and using the same scale.

Line 205 – the word “respectively” should not be abbreviated.

References

The authors lite 125 references but did not cite beyond reference 67 in the main article.

References should ideally list more than the first author. There seems to be some discrepancy here with reference 59 listing up to three authors.

Supplementary files

Table 1 – whilst comprehensive, this table is very user unfriendly. Although it is possible to determine which studies were excluded, probably for an article, would have been to partition the table into studies which were included and those which weren’t or perhaps add an extra column specifying whether included or excluded.

Readability of this article could be improved. Sentences are at times convoluted and probably could be split (e.g. the sentence at line 312). Presentation of the tables within the main article could be rendered more concise.

Author Response

Dear reviewer 2,

We would like to thank you for reviewing our manuscript. Please find attached a word file with our response.

Kind regards.

Round 2

Reviewer 2 Report

I am happy with the responses provided and have nothing further to add